# Reliability and validity of the Perceived Stress Scale in Bangladesh

**Muhammad Kamruzzaman Mozumder** (ORCID) *

Department of Clinical Psychology, University of Dhaka, Dhaka, Bangladesh

* mozumder@du.ac.bd

## Abstract

With validation studies conducted all over the world, the 10-item Perceived Stress Scale (PSS-10) has become a robust and widely used instrument for assessing the appraisal of stress. The present study was aimed at validation and testing for psychometric properties of the PSS-10 on the Bangladeshi population. Three hundred and fifteen adult (aged 18–64 years) from eight divisional districts of Bangladesh participated in this study. A good fit of the two-factor structure of the scale was indicated by multiple indices ($\chi^2/df$, root mean square error of approximation, comparative fit index, and standardized root mean square residual) on confirmatory factor analysis. The full scale demonstrated internal consistency, test-retest reliability and construct validity. The two factors also demonstrated acceptable psychometric properties. The psychometric properties of the Bengali PSS-10 demonstrated in this study suggest the PSS-10 as a valid and reliable instrument for use in Bangladesh and among Bengali-speaking populations.

**Data Availability Statement:** Data available publicly at the Open Science Framework (https://osf.io/dashboard) with Identifier: DOI 10.17605/OSF.IO/A7WME

## Introduction

Stress is intricately connected with both physical and mental aspects of health and well-being, which makes it an important phenomenon for study in psychiatry. The measurement of stress has been traditionally associated with the assessment of stressful life events, in which the presence or the frequency of distinct stressful life events is measured. The importance of an individual's appraisal and its interaction with environmental events in understanding stress was pointed out in the seminal work of Lazarus [1]. This new perspective emphasized the measurement of stress appraisal over the measurement of stressful life events and led Cohen, Kamarck [2] to develop the Perceived Stress Scale (PSS) for assessing perceptions of stress. This appraisal-based scale rapidly gained popularity among researchers because of its unique ability to measure stress appraisal as well as its superior psychometric properties and performance compared to life event-based measures of stress [3]. Over the last three decades, the PSS has been translated into more than 30 languages and has been used in different cultural contexts, including Japan [4], China [5], Turkey [6], Germany [7], Africa [8], and Brazil [9]. Apart from the general population, validation studies on the PSS have also been carried out in different subpopulations, including patients with health problems, psychiatric patients, and different occupational groups.

**Funding:** The author(s) received no specific funding for this work.

**Competing interests:** The authors have declared that no competing interests exist.

The PSS has three versions with 14, 10 and 4 items, among which the 10-item version (i.e., PSS-10) demonstrates superior psychometric properties and is recommended by the original authors [10]. Across different populations, the PSS-10 demonstrates a stable two-factor structure [5, 8, 9]. The first factor has often been termed *perceived helplessness*, *general distress*, or *negative perception*, while the second factor has been termed *perceived self-efficacy*, *ability to cope*, or *positive perception* [4, 6, 11].

The Bengali version of the PSS-10, translated by Ziaul Islam and endorsed by the original authors, is available at the Laboratory for the Study of Stress [12]; however, no data regarding its validity or reliability in general population are provided or is available in the published literature. A recent publication reported adequate psychometric properties of the Bengali PSS-10 on the lesbian, gay, bisexual and transgender (LGBT) population [13]. A review of the literature revealed the availability of two alternative Bengali translations of the PSS-10, neither of which is endorsed by the original authors. A 6-item Bengali version has been tested in India [14]. The Indian scale was found to be internally consistent (Cronbach's alpha = .80); however, due to poor kappa values ($< .5$) of four items, the authors decided to shrink the 10-item scale into a six-item scale, making it non-equivalent to the original 10-item PSS [14]. The translation reliability ($r = .90$, between the Bengali and English version) and the test-retest reliability ($r = .94$) of an unpublished Bengali PSS-10 has also been reported in the literature [see 15].

Despite the lack of published data on psychometric properties, studies have been conducted using Bengali versions of the PSS to assess perceived stress among adolescent girls [16], care-givers of children with autism [17], and adult males [15]. Use of the tool without evidence of psychometric properties in the specified population can cause incorrect use and interpretation of the tool and its findings. Evidence of its psychometric properties is expected to increase confidence in the use of this highly regarded measure of perceived stress in the Bengali-speaking population. The present study, therefore, was designed to assess the reliability, validity and factor structure among the general population in Bangladesh. Based on the available evidence on the psychometric properties of the PSS-10 in other contexts and population groups across the world, including Bangladesh [5, 8, 9, 13, 18], it is hypothesized that the Bengali PSS-10 will demonstrate a two-factor structure as well as adequate reliability and validity in a sample of the Bangladeshi general population.

## Materials and methods

### Participants

A Bengali speaking community sample of 316 incidentally selected adults from all eight divisional districts of Bangladesh participated in this study. The participants were predominantly male (57.3%) and married (61.1%) with age-range of 18 to 64 years ($M = 32.6$, $SD = 10.5$). They varied according to the level of educational attainment (no formal education 11.4%, up to primary 19.6%, up to secondary 15.8%, up to higher secondary 15.8%, up to graduation 22.8%, and above graduation 14.6%). Forty-four participants from the Dhaka district were tested twice, with a gap of two weeks in between, to allow for the assessment of the test-retest reliability of the scale.

### Materials

**The Perceived Stress Scale (PSS-10; [2]).** The ten items are presented with a 5-point response option, providing a score of 0 to 4 for each item and resulting in a score range of 0 to 40 for the total scale. A higher score is indicative of greater perceived stress. There are six forward-scoring items (1, 2, 3, 6, 9 and 10) that constitute the first factor, while the remaining four reverse-scoring items (4, 5, 7 and 8) constitute the second factor. The original PSS-10 and

its different translated versions demonstrated adequate internal consistency, test-retest reliability, and construct validity [10]. The reported internal consistency of the scale indicated by Cronbach's alpha ranged from.71 to .91, while the test-retest reliability has been generally observed at $r > .70$ among different populations [19–21]. Concurrent validity of the PSS-10 was assessed using several tools, such as the depression scale, anxiety scale, the Impact of Event Scale, the General Health Questionnaire (GHQ) and the life event scale, which collectively indicated moderate to strong correlation [10, 20]. The Bengali version of the scale endorsed by the original authors [12] has never been reported in terms of its psychometric properties for a general population sample.

**The General Health Questionnaire (GHQ-28; [22]).** The GHQ-28 is a widely used research instrument with 28 items distributed over four subscales, namely, somatic symptoms, anxiety and insomnia, social withdrawal, and severe depression. Each of the items can have a score of 0 to 3, resulting in a total score range of 0 to 84. Higher scores on the GHQ-28 indicate more distress. The Bengali GHQ-28 used in this study was translated from English by Banoo [as cited in 23], where the translation was evaluated by a panel of 14 judges comprised of psychiatrists and psychologists fluent in both Bengali and English. Adequate test-retest reliability (Spearman's rho = .682) of the scale has been reported [as cited in 23].

**The Self-Reporting Questionnaire (SRQ-20; [24]).** The SRQ-20 was used to assess the overall psychological morbidity among the general population. This instrument consists of 20 items with a dichotomous (no-yes) response option. With the score for the total scale ranging from 0 to 20, a higher score in this scale indicates the presence of more symptoms. Validation studies on the SRQ-20 have been carried out in several countries, including Bangladesh [25], and is widely used as a screening and research tool. Using the score $\geq 8$ as the cutoff value, the SRQ-20 has been reported to have acceptable sensitivity (81%-90%) and specificity (58%-95.2%) in screening people with and without psychological morbidity across populations from different countries, including Italy, Brazil, Nicaragua and Kenya [see 24].

## Procedure

The data used in this article came from two different studies. The first set of data (n = 86) was collected from the general people (heterosexual) who served as a comparison group for a larger study on LGBT mental health. The second set of data (n = 230) was collected from the general population with the specific purpose of validating tools. Data were collected using incidental sampling in the participants' local communities by 14 trained research assistants (RAs) who had academic background in psychology. The RAs walked in the community areas (for example, marketplaces, parks, and housing areas) of the selected cities and invited the individuals they came across to participate in this study. They administered the questionnaire through face-to-face interviews upon verbal consent from participants based on clear and detailed information on the procedure and nature of the study. All data were collected anonymously; however, the participants' mobile phone numbers were collected for the test-retest sample to identify and re-contact them for retesting in two weeks' time. Ethical approvals were received (project ID # IR151101; approved on November 12, 2015 and project ID # IR161101; approved on November 6, 2016) from the relevant ethics committees prior to the commencement of the study.

## Data analysis

All analyses in this study were performed using PASW Statistics 18 [26] and AMOS 18 [27]. No violation was indicated during checks for assumptions of univariate normality (skewness -0.25 to 0.61; kurtosis -0.79 to -0.27) or of univariate outliers (z score -1.23 to 2.33). However,

a check for multivariate outliers using the Mahalanobis distance ($p < .001$) resulted in the removal of one participant, reducing the total number of participants to 315 [28].

The maximum likelihood estimation method was used in the confirmatory factor analysis (CFA) to test the goodness-of-fit of the two-factor model of the PSS-10. Among many different indices used to assess the adequacy of model fit, the chi-square ($\chi^2$), the ratio of chi-square to $df$ ($\chi^2$/df), the root mean square error of approximation (RMSEA), the comparative fit index (CFI), the Tucker-Lewis index (TLI), and the standardized root mean square residual (SRMR) is the most commonly suggested indices and were therefore used in this study. The criteria for model fit were as follows: $\chi^2$ with $p \geq .01$, $\chi^2/df \leq 2$, RMSEA $\leq .06$, CFI $\geq .95$, TLI $\geq .95$, and SRMR $\leq .08$ [29, 30].

Internal consistency reliability was assessed using Cronbach's alpha, and test-retest reliability was assessed using Pearson correlations. Cronbach's alpha $> .70$ was set as the criterion for evaluating internal consistency [31]. The strength of the correlation was determined by Cohen's criteria, where Pearson's $r > .50$ is considered a large correlation [32]. The construct validity of the Bengali PSS-10 was assessed with two convergent methods. First, it was hypothesized that the perception of stress is related to psychological distress, and hence, a positive correlation between the scores on the PSS-10 and the GHQ-28 (a measure of distress) would be indicative of construct validity [see 20]. It was also hypothesized that the two factors of the PSS-10 would have a positive correlation with the GHQ-28 score. It may be noted here that the underlying focuses of the GHQ-28 are the appearance of distressing phenomenon and the inability to carry out normal functions, which logically are connected to the perception of stress measured by the PSS 10. Second, due to the relationship between stress and mental health problems [33, 34], it was hypothesized that the PSS-10 and its factors would be able to discriminate between people with and without mental health problems. With a cutoff value of 8 [24, 25], the SRQ-20 was used to split the sample into two groups: one with mental health problems (SRQ score $\geq 8$) and the other without mental health problems.

## Results

### Confirmatory factor analysis (CFA)

With items 8 and 10 set as the marker variables, the two-factor model was tested through CFA. However, the indices suggested a poor fit of this model (Table 1). Modification indices were then consulted, which suggested a correlation between the error terms for items 6 and 10. A review of the scale items indicated that these two items address the common theme of *being overwhelmed by much work or difficulty*. The Bengali translation of the two items also centered on a single theme, *not being able to do or overcome*. This similarity supported the idea that these two items can share a common error variance. The correlating error terms for these two items have also been tested and suggested in previous studies [35]. Therefore, the original model was slightly modified by correlating the error terms for items 6 and 10 (Fig 1). The resulting model fits well with the data, as indicated by most of the fit indices except for chi-square and TLI (see Table 1 and Fig 1).

**Table 1. Goodness-of-fit indices for the two-factor model of the Bengali PSS-10.**

| | $\chi^2$ | df | p | $\chi^2$/df | RMSEA (CI) | CFI | TLI | SRMR |
|---|---|---|---|---|---|---|---|---|
| Original model | 85.230 | 34 | .000 | 2.507 | .069 (.051-.088) | .916 | .889 | .057 |
| Modified model | 63.850 | 33 | .001 | 1.935 | .055 (.034-.074) | .950 | .931 | .051 |

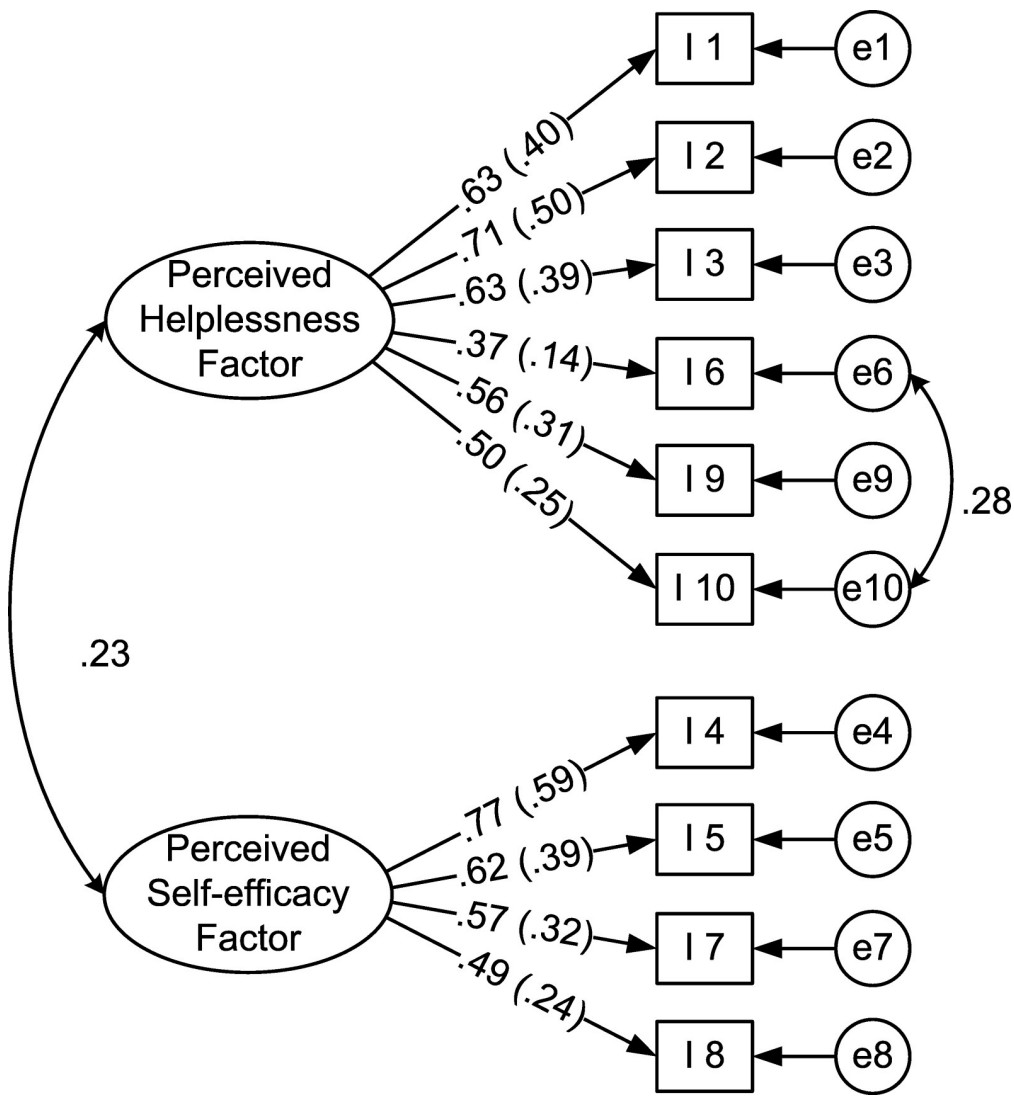

**Fig 1. Modified CFA model of the Bengali PSS-10.** Communalities are presented in the parentheses. I 1—I 10 indicate item numbers, and e1—e10 represent error terms for respective items.

## Internal consistency reliability

Cronbach's alpha was used as an indicator for internal consistency of the scale. For the Bengali PSS-10 full scale alpha was .715, while for the six-item Factor 1 (*perceived helplessness*), the alpha was .750, and for the four-item Factor 2 (p*erceived self-efficacy)*, it was .703.

## Test-retest reliability

The Bengali PSS-10 was administered twice with a two-week interval between administrations on a subsample ($n$ = 44) to assess the test-retest reliability of the scale. A strong correlation ($r$ = .745, $p$ < .01) between the scores from the two administrations indicated test-retest reliability, i.e., stability reliability of the Bengali PSS-10. The test-retest reliabilities of the two factors measured by Pearson correlations were .674 for the *perceived helplessness* factor and .468 for the *perceived self-efficacy* factor.

**Table 2. Mean difference of the PSS-10 scores between individuals with and without mental health problems, as indicated by the SRQ-20.**

|  | Presence of mental health problems | | *t* (313) | *d* |
|---|---|---|---|---|
|  | Yes (*n* = 130) | No (*n* = 185) |  |  |
|  | *M* (SD) | *M* (SD) |  |  |
| Full-Scale PSS-10 | 19.93 (5.78) | 15.39 (5.45) | 7.10* | 0.81 |
| Perceived helplessness factor | 11.61 (4.43) | 8.45 (4.17) | 6.45* | 0.74 |
| Perceived self-efficacy factor | 8.32 (3.20) | 6.95 (3.23) | 3.73* | 0.43 |

* *p* < .01.

### Construct validity

A strong correlation was found between the full-scale PSS-10 score and the GHQ-28 score (*r* = .579, *p* < .01). The *perceived helplessness* factor also had an adequate correlation with the GHQ-28 (*r* = .493, *p* < .01), while the *perceived self-efficacy* factor demonstrated a significant but weak correlation (*r* = .360, *p* < .01).

Independent sample t-tests on the PSS-10 scores between the two groups supported the measure's ability to discriminate between individuals with and without mental health problems (*t* = 7.10, *p* < .01; see Table 2). A large difference [32] was indicated by the effect size of the t-value, calculated using Cohen's d (0.81).

### Discussion

With the aim of assessing the psychometric properties of the Bengali PSS-10 among the general population of Bangladesh, this study analyzed countrywide data from 315 adult participants from eight divisional districts of Bangladesh. The widely reported two-factor structure of the scale on general population across the globe was tested using confirmatory factor analysis. The initial CFA did not indicate an adequate fit of the model. Modification indices suggested a relation between the error terms for items 6 and 10. Subsequent exploration into these two items indicated some overlaps in the item contents (*not being able to do or overcome*). Therefore, the model was slightly modified by correlating the error terms for items 6 and 10 –a solution that has also been proposed by other studies on the PSS-10 factor structure [35]. The modified model (Fig 1) indicated a good fit on multiple fit indices except for chi-square ($\chi^2$) and TLI. The TLI (.931) for the model was very close to the stringent criterion value (TLI ≥ .95) used in this study and hence can be considered indicative of a marginal fit. It is well known that chi-square is highly affected by sample size, and a fit indicated by chi-square criteria is often considered 'difficult to obtain' with a sample size greater than 200, which was the case for the present study (*n* = 315). Therefore, the ratio of chi-square to *df* (1.935), the root mean square error of approximation (.055), the comparative fit index, (.950), the Tucker-Lewis Index (.931) and the standardized root mean square residual (.051) can be considered sufficient evidence of model fit of the two-factor structure of the Bengali PSS-10. The fit of the model implies that the perceived stress indeed has two subconstructs–*perceived helplessness* and *perceived self-efficacy*–that are reflected in the ten items of the PSS-10.

Cronbach's alpha for the full-scale Bengali PSS-10 and its two factors–*perceived helplessness* and *perceived self-efficacy*–were above the acceptable level of .70 [31]. The 6-item *perceived helplessness* factor had the highest Cronbach's alpha (.750). A slightly reduced alpha (.715) for the total scale can be explained by a poor intercorrelation between the two factors (*r* = .154; calculated post hoc). The findings suggested that the PSS-10 and its two factors are internally

consistent. A strong correlation [32] was found between two administrations over a two-week interval for the full scale. The two factors also demonstrated test-retest reliability.

The construct validity of the scale was assessed using two convergent methods. The GHQ-28 was used for the first convergent validation, which is a common practice observed in the PSS-10 validation studies [see 20]. A significant and strong correlation [32] between scores of the PSS-10 and the GHQ-28 confirmed the convergent validity of the scale. The SRQ-20-based screening was used to identify individuals with and without psychological morbidity in the general population sample. A significant difference ($t = 7.10$, $p < .01$) was found between the PSS-10 scores for the two groups, indicating the scale's ability to differentiate between individuals with and without psychological morbidity and adding further evidence of its construct validity. The first factor, that is, *perceived helplessness* also demonstrated strong evidence of its construct validity in both methods. However, the *perceived self-efficacy* factor demonstrated comparatively weaker (although significant) validity in both of the convergent methods.

From the findings presented and discussed here, the Bengali PSS-10 can be deemed a psychometrically sound instrument. Adequate reliability, validity, and evidence of comparable factor structure in the Bangladeshi sample are indicative of the utility of the Bengali PSS-10 in the general Bangladeshi population. Although the lack of randomization and the absence of data on the response rate raise concerns regarding the generalizability of the findings, the inclusion of countrywide participants with different socio-demographic characteristics makes the results useful. The findings also added support to the robustness of the PSS-10 in assessing perceived stress across different cultural contexts.

## Conclusions

Stress is a key construct in understanding health and well-being. It may not be an overstatement to claim that stress is related to all aspects of human living. Thus, it has been an important topic of interest among multidisciplinary researchers. As a fast self-report assessment tool, the PSS-10 has immense research and clinical utility. The findings proved the usability and adequacy of the psychometric properties of the Bengali PSS-10 in assessing perceived stress in the Bangladeshi population. As randomized sampling could not be used, participation from all the divisional headquarters (eight) were ensured as an attempt to ensure representation of the Bangladeshi population in the sample. However, the result can never be generalized and caution should be taken before utilizing and interpreting scale scores from the general population based on current findings. A similar factor structure found for the Bengali PSS-10, and other translated versions of the PSS-10 suggests the comparability of PSS-10 scores across populations of different countries. Evidence of the validity and reliability of the PSS-10 in the Bangladeshi population is likely to increase the use of the scale as a measure of stress appraisal and may substantially contribute to health research in Bangladesh.

## Author Contributions

**Conceptualization:** Muhammad Kamruzzaman Mozumder.

**Formal analysis:** Muhammad Kamruzzaman Mozumder.

**Methodology:** Muhammad Kamruzzaman Mozumder.

**Writing – original draft:** Muhammad Kamruzzaman Mozumder.

**Writing – review & editing:** Muhammad Kamruzzaman Mozumder.

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
