## [Editor Report · Decision Letter 0]

29 Sep 2021

PONE-D-21-21312Reliability and validity of the Perceived Stress Scale in BangladeshPLOS ONE

Dear Dr. Mozumder,

Thank you for submitting your manuscript to PLOS ONE. After careful consideration, we feel that it has merit but does not fully meet PLOS ONE’s publication criteria as it currently stands. Therefore, we invite you to submit a revised version of the manuscript that addresses the points raised during the review process.

We look forward to receiving your revised manuscript.

Kind regards,

Rodrigo Ferrer-Urbina, Ph.D.

Academic Editor

PLOS ONE

2.In your Data Availability statement, you have not specified where the minimal data set underlying the results described in your manuscript can be found. PLOS defines a study's minimal data set as the underlying data used to reach the conclusions drawn in the manuscript and any additional data required to replicate the reported study findings in their entirety. All PLOS journals require that the minimal data set be made fully available. For more information about our data policy, please see http://journals.plos.org/plosone/s/data-availability.

Additional Editor Comments (if provided):

Dear Authors:

First of all, I am grateful for the opportunity to review your manuscript, which seems to me to be a good contribution to the cultural extension of psychometric tests, which is essential for the development of local research and, above all, to contribute to the development of psychology beyond the WEIRD countries. However, I have decided not to send this paper for review, as it has some important flaws that will surely be pointed out by the reviewers, and I am concerned that some of these issues may not be possible to address in the time they would have to do so, so I prefer to return the paper to you so that you can decide whether to make the changes suggested before the review or not to submit it to this journal.

In general terms, it seems to me that the theoretical framework is adequately written, within what is to be expected for a psychometric article, although I believe that the arguments for the relevance of developing validity evidence for this population could be better developed. In addition, there is a lack of theoretical justification of the variables used for the evidence of validity in relation to other variables.

In the methodological aspects, I suggest not using exploratory factor analysis, since there is a well-defined model and, in case it is performed, not using the principal components estimation method, since this estimation is not a factor estimation, since it corresponds to formative (not reflective) models. Another even more important weakness is to perform the exploratory and confirmatory analysis with the same sample, since it is redundant and any spurious structure would be confirmed by mere capitalization of chance (if this double process is done, different samples should be used), producing an overfitting. To avoid this, if sufficient sample size is not available, I suggest considering the application of ESEM, instead of EFA and CFA.

All these aspects mentioned correspond to suggestions that, most probably, would be quickly noticed by the reviewers, but all of them can be corrected relatively easily. What concerns me, given that it requires extending and modifying the sampling, is the sample size, but, above all, the lack of representativeness of the sample, since it claims to refer to the general population, but includes almost a third of the LGBTQ population (with which a particular psychometric study has already been conducted), which is evidenced as a sample recycling that is not consistent with the aims of the study. Although in these disciplines it is very rare to have random samples, it is necessary to give some guarantees that the sample is reasonably close to the population in which the instrument is intended to be used, which is not met here, so it is necessary to conduct additional sampling in order to make the study feasible.

Therefore, I have decided not to send the study for review, unless they complement the sample. Additionally, for the review of psychometric studies, it is required that they make the data available, even if they are only those referring to the items of the instruments (which would not interfere with other subsequent or parallel studies), so that the reviewers can review the adjustments of the measurement models if required.

I wish you much success and hope that my comments can be of some use to you.

Best regards.
---

## [Author Response · Author response to Decision Letter 0]

23 Nov 2021

I have revised the document as per suggestion and have completed the journal requirements suggestion from you. Please find below the response to the additional comments. 

Additional Editor Comments:

1. In general terms, it seems to me that the theoretical framework is adequately written, within what is to be expected for a psychometric article, although I believe that the arguments for the relevance of developing validity evidence for this population could be better developed. In addition, there is a lack of theoretical justification of the variables used for the evidence of validity in relation to other variables.

Response: Thank you for this comment. A new sentence added in introduction. Some details on the justification of the validation variables are included in the last paragraph of data analysis section. 

2. In the methodological aspects, I suggest not using exploratory factor analysis, since there is a well-defined model and, in case it is performed, not using the principal components estimation method, since this estimation is not a factor estimation, since it corresponds to formative (not reflective) models. Another even more important weakness is to perform the exploratory and confirmatory analysis with the same sample, since it is redundant and any spurious structure would be confirmed by mere capitalization of chance (if this double process is done, different samples should be used), producing an overfitting. To avoid this, if sufficient sample size is not available, I suggest considering the application of ESEM, instead of EFA and CFA.

Response: Thank you for this suggestion. I have removed the EFA part of analysis. I strongly agree that this is redundant for this context. 

3. All these aspects mentioned correspond to suggestions that, most probably, would be quickly noticed by the reviewers, but all of them can be corrected relatively easily. What concerns me, given that it requires extending and modifying the sampling, is the sample size, but, above all, the lack of representativeness of the sample, since it claims to refer to the general population, but includes almost a third of the LGBTQ population (with which a particular psychometric study has already been conducted), which is evidenced as a sample recycling that is not consistent with the aims of the study. Although in these disciplines it is very rare to have random samples, it is necessary to give some guarantees that the sample is reasonably close to the population in which the instrument is intended to be used, which is not met here, so it is necessary to conduct additional sampling in order to make the study feasible. 

Therefore, I have decided not to send the study for review, unless they complement the sample.

Response: Please note that ‘No data from LGBT population’ is used in this analysis. All the data are from general population. Part of the data was collected as comparison group for LGBT as no comparable normative data from general population was available at that time. Moreover, none of these data were published in any other articles. Therefore, observation regarding sample recycling is inaccurate. Slight modification of text is made in the method section to reduce confusion. 

I agree completely that this is not at all a complete representation of general population of the country. However, we try tried to get closer to that by collecting data from all the Eight divisions of Bangladesh. Achieving more that that would require huge resources reducing feasibility of the study. A sentence has been added to the conclusion section. 

4. Additionally, for the review of psychometric studies, it is required that they make the data available, even if they are only those referring to the items of the instruments (which would not interfere with other subsequent or parallel studies), so that the reviewers can review the adjustments of the measurement models if required.

Response: Thank you for making it clearer, now I have made the data publicly available.

---

## [Decision Letter · Decision Letter 1]

7 Jul 2022

PONE-D-21-21312R1Reliability and validity of the Perceived Stress Scale in BangladeshPLOS ONE

Dear Dr. Mozumder,

Thank you for submitting your manuscript to PLOS ONE. After careful consideration, we feel that it has merit but does not fully meet PLOS ONE’s publication criteria as it currently stands. Therefore, we invite you to submit a revised version of the manuscript that addresses the points raised during the review process. Please note that we have only been able to secure a single reviewer to assess your manuscript. We are issuing a decision on your manuscript at this point to prevent further delays in the evaluation of your manuscript. Please be aware that the editor who handles your revised manuscript might find it necessary to invite additional reviewers to assess this work once the revised manuscript is submitted. However, we will aim to proceed on the basis of this single review if possible. The reviewer has raised a number of minor concerns that need attention. They request additional information and/ or rewording on methodological aspects of the study. Could you please revise the manuscript to carefully address the concerns raised? Please submit your revised manuscript by Aug 21 2022 11:59PM. If you will need more time than this to complete your revisions, please reply to this message or contact the journal office at plosone@plos.org. Please include the following items when submitting your revised manuscript:A rebuttal letter that responds to each point raised by the academic editor and reviewer(s). You should upload this letter as a separate file labeled 'Response to Reviewers'.A marked-up copy of your manuscript that highlights changes made to the original version. You should upload this as a separate file labeled 'Revised Manuscript with Track Changes'.An unmarked version of your revised paper without tracked changes. You should upload this as a separate file labeled 'Manuscript'.If applicable, we recommend that you deposit your laboratory protocols in protocols.io to enhance the reproducibility of your results. Protocols.io assigns your protocol its own identifier (DOI) so that it can be cited independently in the future. For instructions see: https://journals.plos.org/plosone/s/submission-guidelines#loc-laboratory-protocols. Additionally, PLOS ONE offers an option for publishing peer-reviewed Lab Protocol articles, which describe protocols hosted on protocols.io. Read more information on sharing protocols at https://plos.org/protocols?utm_medium=editorial-email&utm_source=authorletters&utm_campaign=protocols.

We look forward to receiving your revised manuscript.

Kind regards,

Katrien Janin

Staff Editor

PLOS ONE

Journal Requirements:

Reviewers' comments:

Reviewer's Responses to Questions

**Comments to the Author**

1. If the authors have adequately addressed your comments raised in a previous round of review and you feel that this manuscript is now acceptable for publication, you may indicate that here to bypass the “Comments to the Author” section, enter your conflict of interest statement in the “Confidential to Editor” section, and submit your "Accept" recommendation.

Reviewer #1: (No Response)

2. Is the manuscript technically sound, and do the data support the conclusions?

Reviewer #1: Yes

3. Has the statistical analysis been performed appropriately and rigorously? 

Reviewer #1: Yes

4. Have the authors made all data underlying the findings in their manuscript fully available?

Reviewer #1: Yes

5. Is the manuscript presented in an intelligible fashion and written in standard English?

Reviewer #1: Yes

6. Review Comments to the Author

Reviewer #1: It seems to me that the paper presented is generally well written and analyzes the psychometric properties of the scale in a sample of Bangladeshi people. However, there are some details that I think can be improved.

I think it would be important to explain better, why theoretically what the PSS-10 measures should be related to what the GHQ-28 measures, so that the reader has more clarity regarding the nomological network around the PSS-10.

Also, it would be advisable to include the operational definition of the factors represented by the PSS-10, in order to have a better understanding of the meaning of the results obtained.

On the other hand, there are some assertions in the paper that could be misinterpreted. For example, in the instruments part it says "The SRQ-20 has been validated in several countries, including Bangladesh [25], and is widely used as a screening and research tool", this could lead to believe that "the tests are valid", however, as explained in more detail by the AERA, APA, & NCME (2014), the tests are not valid, but the use and interpretation of the scores could be or not.

A little more precision is needed regarding the wording of the results obtained from Cronbach's Alpha coefficient. I recommend reading Cho and Kim (2015) or Hoekstra et al., (2018), where they explain some common errors in the interpretation of this coefficient.

Finally, taking into consideration that Bangladesh is a country with more than 150 million inhabitants and that the presented study does not have randomized sampling, I recommend being more careful about generalizing the results.

American Educational Research Association, American Psychological Association, & National Council on Measurement in Education (2014). Standards for educationaland psychological testing. Washington, DC: American Educational Research Association. https://www.testingstandards.net/open-access-files.html

Cho, E., & Kim, S. (2015). Cronbach’s Coefficient Alpha: Well Known but Poorly Understood. Organizational Research Methods, 18(2), 207–230. https://doi.org/10.1177/1094428114555994

Hoekstra, R., Vugteveen, J., Warrens, M. J., & Kruyen, P. M. (2019). An empirical analysis of alleged misunderstandings of coefficient alpha. International Journal of Social Research Methodology, 22(4), 351-364. https://doi.org/10.1080/13645579.2018.1547523

7. PLOS authors have the option to publish the peer review history of their article (what does this mean?). If published, this will include your full peer review and any attached files.

Reviewer #1: No

---

## [Author Response · Author response to Decision Letter 1]

4 Aug 2022

Responses to the Reviewer’s comments 

Thank you for reviewing this manuscript and making suggestion to improve the manuscript (PONE-D-21-21312R1; Reliability and validity of the Perceived Stress Scale in Bangladesh). Please find my responses after each of your suggestions (listed in numbers). 

1. I think it would be important to explain better, why theoretically what the PSS-10 measures should be related to what the GHQ-28 measures, so that the reader has more clarity regarding the nomological network around the PSS-10.

Response: Thank you for this suggestion, I have added a sentence in the manuscript to address this, “It may be noted here that the underlying focuses of the GHQ-28 are the appearance of distressing phenomenon and the inability to carry out normal functions, which logically are connected to the perception of stress measured by the PSS 10.” (line 151-153, page 8)

2. Also, it would be advisable to include the operational definition of the factors represented by the PSS-10, in order to have a better understanding of the meaning of the results obtained.

Response: I agree that this would be useful however, I did not find any operational definition from the authors who first coined these two factors. 

3. On the other hand, there are some assertions in the paper that could be misinterpreted. For example, in the instruments part it says "The SRQ-20 has been validated in several countries, including Bangladesh [25], and is widely used as a screening and research tool", this could lead to believe that "the tests are valid", however, as explained in more detail by the AERA, APA, & NCME (2014), the tests are not valid, but the use and interpretation of the scores could be or not.

Response: This was a new insight for me, thank you for your comment. The sentence has now been revised to avoid the misinterpretations, “Validation studies on the SRQ-20 have been carried out in several countries, including Bangladesh”. (line 103-104, page 6)

I have also revised another sentences based on this suggestion, “validation studies on the PSS have also been carried out in different subpopulations” (line 29-30, page 3)

4. A little more precision is needed regarding the wording of the results obtained from Cronbach's Alpha coefficient. I recommend reading Cho and Kim (2015) or Hoekstra et al., (2018), where they explain some common errors in the interpretation of this coefficient.

Response: This is very useful suggestion. Thank you for the recommended articles, they explained much more details on the misinterpretations of Cronbach's Alpha. I have rephrased some sentences to avoid such misinterpretations.

“The reported internal consistency of the scale indicated by Cronbach’s alpha ranged from.71 to .91” (line 83-84, page 5)

“Cronbach’s alpha was used as an indicator for internal consistency of the scale. For the Bengali PSS-10 full scale alpha was .715” (line 180-181, page 9)

The word “proved” has been replaced with “suggested”, “The findings suggested that the PSS-10 and its two factors are internally consistent” (line 231, page 12)

5. Finally, taking into consideration that Bangladesh is a country with more than 150 million inhabitants and that the presented study does not have randomized sampling, I recommend being more careful about generalizing the results.

Response: Although a caution was reflected at the end of discussion (“Although the lack of randomization and the absence of data on the response rate raise concerns regarding the generalizability of the findings, the inclusion of countrywide participants with different socio-demographic characteristics makes the results useful.”; line 251-254, page 12) a new sentence has now been added in the conclusion section, “However, the result can never be generalized and caution should be taken before utilizing and interpreting scale scores from the general population based on current findings.” (line 264-266, page 13)

Additionally, in the abstract the word “established” has been replaced with “suggest” in sentence “this study suggest the PSS-10 as a valid and reliable instrument“ (line 11-12, page 2)

---

## [Decision Letter · Decision Letter 2]

17 Oct 2022

Reliability and validity of the Perceived Stress Scale in Bangladesh

PONE-D-21-21312R2

Dear Dr. Mozumder,

We’re pleased to inform you that your manuscript has been judged scientifically suitable for publication and will be formally accepted for publication once it meets all outstanding technical requirements.

Kind regards,

Jeffrey S. Hallam, Ph.D., FRSPH

Academic Editor

PLOS ONE

Additional Editor Comments (optional):

Reviewers' comments:

Reviewer's Responses to Questions

**Comments to the Author**

1. If the authors have adequately addressed your comments raised in a previous round of review and you feel that this manuscript is now acceptable for publication, you may indicate that here to bypass the “Comments to the Author” section, enter your conflict of interest statement in the “Confidential to Editor” section, and submit your "Accept" recommendation.

Reviewer #1: All comments have been addressed

Reviewer #2: (No Response)

2. Is the manuscript technically sound, and do the data support the conclusions?

Reviewer #1: Yes

Reviewer #2: Yes

3. Has the statistical analysis been performed appropriately and rigorously? 

Reviewer #1: Yes

Reviewer #2: Yes

4. Have the authors made all data underlying the findings in their manuscript fully available?

Reviewer #1: Yes

Reviewer #2: Yes

5. Is the manuscript presented in an intelligible fashion and written in standard English?

Reviewer #1: Yes

Reviewer #2: Yes

6. Review Comments to the Author

Reviewer #1: (No Response)

Reviewer #2: abstract need to be clear in terms of the main research questions. The author should rewrite the abstract section to support the study purposes with each hypothesis ideas and major finding clearly. Add more details in the limitation part.

7. PLOS authors have the option to publish the peer review history of their article (what does this mean?). If published, this will include your full peer review and any attached files.

Reviewer #1: No

Reviewer #2: **Yes: **Reham Bakhsh

---

## [Editor Report · Acceptance letter]

20 Oct 2022

PONE-D-21-21312R2 

Reliability and validity of the Perceived Stress Scale in Bangladesh 

Dear Dr. Mozumder:

I'm pleased to inform you that your manuscript has been deemed suitable for publication in PLOS ONE. Congratulations! Your manuscript is now with our production department. 

Kind regards, 

on behalf of

Dr. Jeffrey S. Hallam 

Academic Editor

PLOS ONE